# Growth Hormone (GH) Therapy During the Transition Period: Should We Think about Early Retesting in Patients with Idiopathic and Isolated GH Deficiency?

**DOI:** 10.3390/ijerph16030307

**Published:** 2019-01-23

**Authors:** Laura Penta, Marta Cofini, Laura Lucchetti, Letizia Zenzeri, Alberto Leonardi, Lucia Lanciotti, Daniela Galeazzi, Alberto Verrotti, Susanna Esposito

**Affiliations:** 1Paediatric Clinic, Department of Surgical and Biomedical Sciences, Università degli Studi di Perugia, 06132 Perugia, Italy; laura.penta@ospedale.perugia.it (L.P.); marta.cofini@gmail.com (M.C.); lauralucchetti@yahoo.it (L.L.); zenzeriletizia@gmail.com (L.Z.); alberto.leonardi88@gmail.com (A.L.); lucia.lanciotti@gmail.com (L.L.); 2Paediatric Unit, Narni Hospital, 05035 Narni (PG), Italy; Daniela.galeazzi@uslumbria2.it; 3Department of Paediatrics, University of L’Aquila, 67100 L’Aquila, Italy; alberto.verrottidipianella@univaq.it

**Keywords:** growth hormone deficiency, transition age, growth hormone retesting

## Abstract

To investigate growth hormone (GH) secretion at the transition age, retesting of all subjects who have undergone GH replacement therapy is recommended when linear growth and pubertal development are complete to distinguish between transitional and persistent GH deficiency (GHD). Early retesting of children with idiopathic and isolated GHD (i.e., before the achievement of final height and/or the adult pubertal stage) can avoid possible over-treatment. Here, we report data from our population with idiopathic and isolated GHD to encourage changes in the management and timing of retesting. We recruited 31 patients (19 males) with idiopathic GHD who received recombinant GH (rGH) for at least 2 years. All of the patients were retested at the transition age at least 3 months after rGH discontinuation. Permanent GHD was defined as a GH peak of <19 ng/mL after administration of growth hormone–releasing hormone (GHRH) + arginine as a provocative test. Permanent GHD was confirmed in only five of 31 patients (16.13%). None of these patients presented low serum insulin-like growth factor (IGF)-1 levels (<−2 standard deviation score (SDS)). Only one male patient with an IGF-1 serum level lower than −2 SDS showed a normal GH stimulation response, with a GH peak of 44.99 ng/mL. Few patients with idiopathic and isolated GHD demonstrated persistence of the deficit when retested at the transition age, suggesting that the timing of retesting should be anticipated to avoid overtreatment.

## 1. Background

The diagnosis of growth hormone deficit (GHD) during childhood represents a complex process that includes auxological and clinical parameters and is associated with neuroradiological and chemical assessments [1]. Idiopathic GHD is defined as either isolated or with one additional hormone deficit in the absence of a defined genetic cause, structural hypothalamic–pituitary abnormalities, central nervous system tumours or a history of high-dose cranial irradiation [2]. Growth hormone (GH) replacement therapy is a cornerstone to guarantee correct growth during childhood, but it also plays a significant role in maintaining the physiological body composition and metabolic homeostasis throughout life [3]. 

After childhood, linear growth is mostly completed, but significant changes in body composition still occur before adulthood; for example, the bone mass peak is achieved, and somatic development continues [2,4]. This life period is defined as the “transition age” and extends from late puberty to complete adult maturation, which usually ranges from the late teens until 6–7 years after the attainment of final height [2,5,6]. Low or undetectable GH levels during the transition and adult ages have been associated with alterations in the somatic composition, altered body development, an increased free-fat mass with increased lipid blood levels, a compromised peak of bone mass and a reduction in muscular strength [7,8,9,10,11]. Therefore, identifying patients with persistent GHD is mandatory because the majority of these alterations can be prevented or widely reduced by correct GH replacement therapy [2,11,12,13,14,15,16,17,18,19,20,21,22,23,24].

To investigate GH secretion at the transition age, the European Society of Paediatric Endocrinology (ESPE) recommends retesting all subjects who have undergone GH replacement therapy when linear growth and pubertal development are complete to distinguish between transitional and persistent GHD [2,25]. Retesting should be performed at least one month after GH discontinuation to correctly determine GH physiological secretion [2]. The importance of restarting replacement therapy in cases with persistent GHD is well established, but data from the literature have reported persistent GHD in only a small percentage of children with childhood-onset GHD. Furthermore, patients formerly diagnosed with idiopathic and isolated GHD have a low likelihood of presenting with persistent GHD during the transition age and adulthood [2]. Therefore, early retesting of children with idiopathic GHD (i.e., before the achievement of final height and/or the adult pubertal stage) can avoid possible over-treatment. We investigated our patients formerly diagnosed with idiopathic and isolated GHD to better understand metabolic features and the real percentage of transient and permanent GHD. Then, we reported data from our population with idiopathic and isolated GHD to encourage changes in the management and timing of retesting.

## 2. Materials and Methods

### 2.1. Patients and Methods

Thirty-one patients (19 males) regularly followed at the Outpatient Clinic for Paediatric Endocrinology in Perugia, Italy, were recruited. All of the patients had a diagnosis of idiopathic GHD during childhood and were retested at the transition age according to the Consensus statement on the management of GH-treated adolescents from the transition to adult care [2]. 

The body weight was measured using a standard balance with a variability of 0.1 kg, and height was determined with a standard stadiometer with variability of 0.1 cm. The body mass index (BMI) was calculated as the ratio between weight expressed in kg and height expressed in square metres (kg/m^2^) and then converted to a standard deviation score (SDS) using Italian reference data [17]. The mid-parental height (MPH) expressed in cm and then reported in correspondent SDS values and calculated according to Tanner’s method as follow: Boys: (father’s height + 13 + mother’s height)/2; Girls (father’s height − 13 + mother’s height)/2 [26]. The pubertal stage was assessed according to the Tanner stages for females with breast development [26] and for males with the volume of both testes checked with standard Prader’s orchidometer, considering >20 mL as stage V [24,27,28].

At baseline, the height, weight, BMI and MPH were evaluated in all subjects, and personal and family histories were investigated. To exclude the presence of multiple pituitary deficiencies, thyroid stimulating hormone, free thyroxine, adrenocorticotropic hormone, cortisol, follicle-stimulating hormone, luteinizing hormone and gonadal steroid tests were performed. Furthermore, karyotype analysis was performed in all females to exclude Turner syndrome.

At diagnosis, GH secretion was assessed using clonidine (100 µg/m^2^ orally), arginine (0.5 g/kg intravenously) or GH releasing hormone (GHRH, 1 µg/kg intravenously) + arginine (0.5 g/kg intravenously) as secretagogues. In each test, blood samples for GH measurements were obtained at +0, +30, +60, +90 and +120 min after administration of the stimulus. According to current indications of the Italian Medicine Agency, a subnormal response was considered if the GH blood levels were <20 ng/mL when the GHRH + arginine provocative test was administered, whereas a GH blood peak response < 8 ng/mL was considered as a cut-off when the clonidine or arginine provocative test were performed (before 2014, a cut-off < 10 ng/mL was indicated by the Italian Medicine Agency) [28,29].

Brain magnetic resonance imaging (MRI) was performed at diagnosis in all subjects, and the presence of major alterations, especially midline congenital malformations involving the sellar or suprasellar region, was considered an exclusion criterion [30,31,32]. Other exclusion criteria included a history of spinal or total body irradiation, bone dysplasia, significant systemic illness and multiple pituitary hormone deficiency.

Instead, according to previous studies, we considered as minor and/or non-specific findings small pituitary gland, thin stalk, partial asymmetry of the pituitary gland and partial empty sella [30,31,32,33,34]. 

Idiopathic GHD was defined as subnormal responses to two different stimulation tests associated with normal brain neuroimaging or minor/nonspecific findings on MRI.

All eligible patients received recombinant GH (rGH) treatment for 2 or more years. Therapy was stopped when linear growth was mostly completed with a height velocity under 2 cm/year and/or pubertal maturation defined as Tanner stage V. Furthermore, some patients were retested for poor adherence to GH therapy. All subjects were retested at least 3 months after the end of treatment. 

At the retesting time point, the GHRH + arginine stimulation test was performed to evaluate the persistence of GHD using a cut-off of <19 ng/mL according to the Italian Medicine Agency [28,29]. Furthermore, the serum total cholesterol, low-density lipoprotein (LDL) cholesterol, high-density lipoprotein (HDL) cholesterol, triglyceride, blood glucose, and insulin levels and the Homeostasis Model Assessment-estimated Insulin Resistance (HOMA-IR) index were checked to investigate lipid and glucose metabolism. For the HOMA-IR index, a value < 3 was considered a cut-off for pubertal age. 

Based on the “Consensus Statement on the management of the GH-treated adolescent in the transition to adult care”, our cohort could be classified as having a low likelihood of permanent GHD. Indeed, this group included patients with idiopathic GHD that was either isolated or concomitant with one additional hormone deficit [2]. 

### 2.2. Laboratory Procedures

The serum insulin-like growth factor 1 (IGF-1) level was detected using a double solid-phase chemiluminescent immunometric assay. The IGF-1 concentrations were converted to SDS using reference ranges standardized for age and sex. GH was measured using a simultaneous immuno-enzymatic assay. 

### 2.3. Statistical Analyses

The data are shown as the mean ± SDS if not differently indicated. The statistical analyses were performed with the SPSS software (IBM SPSS Statistics for Windows, Version 21.0. Armonk, NY, USA: IBM Corp). A *p* value < 0.05 was considered significant.

### 2.4. Ethical Statement

All subjects gave their informed consent for inclusion before they participated in the study. The study was conducted in accordance with the Declaration of Helsinki, and the protocol was approved by the Ethics Committee of Umbria Region, Perugia, Italy (2015-PED-03).

## 3. Results

Table 1 shows the study population data. The children’s mean age at the start of GH therapy for CO-GHD was 10.7 ± 2.87 years (age range, 3.34–14.44). At the initial diagnosis, all patients demonstrated a subnormal response to two different provocative tests. None of the children presented other additional hormone deficiencies.

Sixteen patients (51.61%) showed no alterations of the hypothalamic-pituitary area on the MRI scan, whereas 15 patients (48.39%) presented minor/non-specific alterations (small pituitary gland, slight asymmetry of pituitary gland, thin stalk and partial empty sella). No midline structural alterations were found in any patients on the MRI. 

All patients were treated with rGH for at least 2 years, with a mean duration of GH therapy of 5.27 ± 2.54 years (2–13). The mean age at retesting was 15.95 ± 1.20 years (range 12.1–17.85) with a mean GH-free wash out period of 0.41 ± 0.12 years (0.25–0.66). The mean age for females was 15.4 ± 1.4 years, while it was 16.3 ± 0.9 years for males.

A GHRH + arginine stimulation test was performed in all patients. The mean peak GH response to the test was 37.43 ± 19.05 ng/mL (0.44–96.64). Five patients (16.13%) had a GH peak < 19 ng/mL; of them, only one (male) showed a very low response to the stimulation test with a GH peak of 0.44 ng/mL.

None of the patients presented low serum IGF-1 levels (<−2 SDS); the only male patient with an IGF-1 serum level lower than −2 SDS showed a normal GH stimulation response with a GH peak of 44.99 ng/mL. In this patient, no alterations were found on the MRI; furthermore, his BMI was 17.27 (−1.64 SDS).

Only one male patient of the five subjects with a pathologic GH peak also showed an alteration on the brain MRI at diagnosis, in particular the neuroradiologic assessment at diagnosis showed presence of partly empty saddle. The IGF-1 level was in the normal range, and his BMI was 22.32.

Twenty-six patients (83.87%) who presented normal IGF-1 concentrations and a normal response to the GHRH + arginine test were discharged from follow-up for a reduced risk of evolving into endocrinopathy, whereas endocrine follow-up was recommended for the five patients with subnormal GH stimulation.

No significant difference in the GH peak was found between the children with a normal MRI and those with minor/non-specific alterations.

The mean BMI value of our patients was 20.51 ± 2.94 (15.73–26.78), −0.48 ± 1.13 SDS (−2.98/+1.56). None of the patients had a BMI > 30, although four patients had BMI values ranging from 25 to 30. 

Regarding lipid profile alterations, we reported one female patient with a BMI of 26.78 associated with increased total cholesterol and a subnormal GH peak at retesting but a normal IGF-1 level. High total cholesterol and LDL cholesterol levels associated with a normal BMI, a normal response to GH retesting and a normal serum IGF-1 level were found in one other female. One male subject with persistent GHD showed a low HDL level; his BMI was 18.51, and he had a normal IGF-1 level. Finally, one female subject had slightly increased total cholesterol, but no alterations in the GH peak or IGF-1 level were found, and her BMI was 20.40. None of the patients presented alterations in the glucose profile; in particular, none of the patients showed insulin resistance evaluated by the HOMA-IR index.

## 4. Discussion

The results of our study demonstrated that the majority of subjects treated with rGH during childhood for idiopathic and isolated GHD, showed a complete recovery of normal GH secretion when retested at the transition age. These data agree with the results of many previously published studies [15,19,21,22,23,24,33,35]. The potential causes of GH response normalization could include real transient GHD, physiological improvement of hypothalamic-pituitary functions after puberty, neuro-secretory dysfunctions with a normal response to the provocative test but altered spontaneous release of GH, several changes in diagnostic criteria, or poor reproducibility of the GH provocative test [36]. The choice of provocative test, the anthropometric parameters of the investigated subjects, such as BMI, and/or an association of other pituitary hormone deficiencies could also play a role [36]. 

Although the insulin tolerance test (ITT) is recommended for retesting because it investigates both GH secretion and hypothalamus-pituitary-adrenal axis function [37,38], this test can induce severe side effects and is contraindicated in some patients [2,39]; therefore, the glucagon test and GH-RH + arginine test may be useful alternatives [2]. We chose the GH-RH + arginine test because it demonstrated a high specificity (100%) and high sensitivity (97%) [33,40], it was more reproducible and it presented fewer adverse effects [34,41,42,43]. 

Unfortunately, the exact cut-off value at retesting is still a subject of debate. We used a cut-off value of <19 ng/mL to define a subnormal response according to the Italian medicine agency [28]. Instead, Dreismann et al. detected 15.9 ng/mL as the highest sensitivity and specificity cut-off [43]. Thus, although GHRH + arginine remains a good provocative test, standardization is needed. Adjusting our results with Dreismann’s cut-off, we would have found four instead of five patients with persistent GHD. 

Currently, there is not strong evidence or a wide consensus about priming for GH test stimulation at diagnosis Furthermore, there are not standardized protocols for priming administration both in male and in female subjects [1,44]. Then, we choose to not perform priming in peripuberal children because this technique is still not standardized, seems only to slightly increase GH response and may underreport peripuberal children that would really benefit from rGH therapy [44]. 

Several studies have been conducted on different subjects, especially in terms of BMI. The negative influence of over-weight and obesity on the GH peak in the provocative test is well established. Therefore, appropriate cut-offs should also consider the BMI values [45]. Different cut-off limits have been proposed according to different BMI ranges over the years; although data are still conflicting, the majority consider <4 ng/mL a reliable cut-off to define a subnormal GH peak at retesting for obese subjects with a BMI >30 [1,33]. In our cohort, none of the subjects had a BMI > 30, but four subjects had a BMI > 25; of these four subjects, only one female also showed a subnormal response at retesting with a GH peak < 19 ng/mL.

According to the Consensus statement on management of the GH-treated adolescent in the transition to adult care, our cohort of GHD patients could be defined as having a low likelihood of permanent GHD due to the absence of multiple hormonal deficiencies and/or major alterations on MRI; therefore, in our population, the possibility of transient GHD is increased [2]. Bizzarri et al. performed early retesting in 38 children formerly diagnosed with idiopathic GHD before the transition age; a total of 95% of the retested subjects showed complete normalization of GH secretion without the need to resume rGH therapy [1].

In a more recent study Vuralli et al. have retested all patients formerly diagnosed with GHD after only 1 year of treatment: the 40.6% of those with isolated GHD had a complete normalization of GH secretion, even if they had showed a severe deficiency with a very low GH-peak at diagnosis [46].

## 5. Conclusions

In conclusion, few patients with idiopathic and isolated GHD demonstrated persistence of the deficit when they were retested at the transition age, suggesting that children with idiopathic and isolated GHD should be retested before achieving their final height and/or adult pubertal stage to avoid over-treatment. On the other hand, after GH therapy discontinuation, endocrinologic pediatric follow-up should still continue until the transition age to assess pubertal and statural development. A review and a standardization of the diagnostic criteria for the diagnosis of GHD is needed because improving the diagnosis would help itself to avoid over-treatment. 

## Figures and Tables

**Table 1 ijerph-16-00307-t001:** Characteristics of children with a diagnosis of idiopathic and isolated GHD during childhood who were retested at the transition age.

Subject	Sex	Age at Diagnosis (Year)	MPH (SDS)	Puberal Stage	MRI	Treatment Period (Year)	Time Off-Treatment (Year)	Age at Retesting	FH (SDS)	Δ FH-MPH (SDS)	GH-Peak (ng/mL)	IGF-1 (ng/mL)	BMI	BMI (SDS)	Glucid Profile	Lipid Profile
1	F	12.70	−0.38	puberal	small pituitary gland	3.5	0.5	16.90	−0.51	−0.13	32.95	237	22.10	0.35	N	N
2	F	11.33	−1.08	puberal	slightly reduced neurohyp signal.	4	0.33	15.84	−0.81	0.27	46.75	407	23.10	0.69	N	N
3	M	5.33	−1	pre-puberal	small pituitary gland	9.33	0.25	15.04	−0.03	0.97	38.74	492	19.65	−0.58	N	N
4	M	10.20	−0.24	pre-puberal	normal	4.5	0.42	15.11	0.1	0.34	15.45	861	18.68	−0.97	N	N
5	M	11.39	−0.85	pre-puberal	normal	5	0.25	17.32	−0.97	−0.12	13.79	432	18.51	−1.44	N	HDL↓
6	M	13.24	−0.68	puberal	normal	3	0.66	17	−0.77	−0.09	43.26	371	15.73	−2.98	N	N
7	M	11.34	−1.05	pre-puberal	thin stalk	4.42	0.42	16.57	−2.53	−1.48	28.24	478	17.89	−1.62	N	N
8	F	12.70	−1.08	puberal	normal	3.42	0.58	16.70	−1.52	−0.44	35.87	586	17.32	−1.81	N	N
9	M	9.90	−0.84	pre-puberal	normal	5.5	0.42	14.82	0.35	1.19	24.74	390	20.21	−0.34	N	N
10	M	6.85	−1.47	pre-puberal	partial empty saddle	8.92	0.33	16.20	−1.68	−0.21	59.02	317	19.43	−0.86	N	N
11	F	4.20	−1.89	pre-puberal	normal	7.42	0.50	12.10	−1.14	0.75	25.94	437	20.59	0.22	N	TG and LDL↑
12	F	9.22	−0.44	pre-puberal	normal	4.5	0.50	14.20	−0.35	0.09	66.43	364	16.76	−1.79	N	N
13	M	12.35	−0.85	pre-puberal	small pituitary gland	3	0.66	16.21	−1.45	−0.6	40.51	349	19.78	−0.72	N	N
14	M	10	−1.08	pre-puberal	small pituitary gland	5.5	0.66	16.20	−0.08	1	35.30	440	19.27	−0.92	N	N
15	M	13.92	−0.90	puberal	slight-asymmetry of pituitary gland	2	0.33	16.41	−0.95	−0.05	42.27	443	20.22	−0.97	N	N
16	F	9.97	−0.44	pre-puberal	small pituitary gland	5.42	0.33	16.80	−0.29	0.15	50.49	490	25.62	1.32	N	N
17	M	8.67	−0.46	pre-puberal	normal	4.5	0.25	14.90	−0.33	0.13	34.40	353	22.05	0.25	N	N
18	F	9.64	0.92	pre-puberal	small pituitary gland	4.33	0.42	14.83	0.10	−0.82	36.30	494	17.23	−1.64	N	N
19	M	14.44	−0.85	puberal	partial empty saddle	2.5	0.58	17.85	−0.46	0.39	0.44	460	22.32	0.11	N	N
20	F	9.79	−1.38	pre-puberal	small pituitary gland	5	0.33	15.31	−2.79	−1.41	58.63	498	26.45	1.50	N	N
21	M	11.64	−0.93	pre-puberal	normal	5.5	0.25	17.40	−0.55	0.38	45.54	407	20.09	−0.72	N	N
22	M	6.22	0.41	pre-puberal	normal	8.92	0.42	15.73	−0.61	−1.02	44.99	173	17.27	−1.64	N	N
23	F	11.11	−0.68	pre-puberal	small pituitary gland	3	0.42	14.95	−1.38	−0.7	96.64	730	20.40	−0.22	N	N
24	M	12.95	−1.71	pre-puberal	normal	3.5	0.33	16.92	−0.98	0.73	35.30	379	19.10	−1.10	N	N
25	F	11.71	−0.85	puberal	normal	4	0.50	16.64	−1.27	−0.42	4.64	269	26.78	1.56	N	TC↑
26	F	7.74	−0.96	pre-puberal	normal	7.5	0.33	15.7	−0.71	0.25	55.64	286	21.37	0.13	N	N
27	F	10.66	−1.12	puberal	partial empty saddle	3.5	0.42	16.64	−1.13	−0.01	25.46	386	22.01	0.36	N	N
28	M	5.47	−0.77	pre-puberal	normal	10.2	0.33	15.94	−0.06	0.71	16.68	453	16.35	−2.30	N	N
29	M	3.34	−0.86	pre-puberal	thin stalk	13	0.42	15.70	−0.64	0.22	39.32	287	21.44	−0.04	N	N
30	M	12	−1.51	pre-puberal	normal	4.75	0.33	17.33	−0.71	0.8	46.21	444	22.23	0.09	N	N
31	M	12.17	−1.63	pre-puberal	normal	3.6	0.33	17.17	−1.89	−0.26	20.40	272	25.97	1.17	N	N

HDL, cholesterol high-density lipoprotein; LDL, cholesterol low-density lipoprotein; N, normal; TG, triglycerides.

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
