# Peer review of "Growth Hormone (GH) Therapy During the Transition Period: Should We Think about Early Retesting in Patients with Idiopathic and Isolated GH Deficiency?"

_ijerph, 2019, doi:10.3390/ijerph16030307_

Round 1
Reviewer 1 Report
Penta, et al, propose early retesting in children with idiopathic GHD.
Repeat GH stimulation testing is recommended at completion of growth for children with GHD.
The article is well-written and the authors are investigating an important and relevant question. The authors present a clinical trial documenting the retesting of children with idiopathic GHD during the transition period. Their results are consistent with previous publications with few individuals failing the GH stimulation test.
Based upon their findings, the authors conclude that “children with idiopathic GHD should be retested before achieving their final height and/or adult pubertal stage to avoid over-treatment. Moreover, performing the GH retest before the transition age allows paediatricians to identify permanent GHD at an early stage and consequently to establish better follow-up before the transition to endocrinological adult care.”
Although the authors provide a recommendation to test earlier, they don’t provide evidence that earlier testing period will be valid or more detail on the timing of earlier testing that they recommend. The authors should provide this information as there are previous publications suggesting earlier testing and early discontinuation of therapy.
The authors use a stimulation test that is well-validated, but unfortunately not available to everyone. In particular, GHRH is unavailable in the US. This makes the results of this study difficult to generalize.
Author Response
….Although the authors provide a recommendation to test earlier, they don’t provide evidence that earlier testing period will be valid or more detail on the timing of earlier testing that they recommend. The authors should provide this information as there are previous publications suggesting earlier testing and early discontinuation of therapy.
o The sentence was clarified as suggested.
The authors use a stimulation test that is well-validated, but unfortunately not available to everyone. In particular, GHRH is unavailable in the US. This makes the results of this study difficult to generalize.
We chose the GH-RH + Arginine test because of it is more useful (reproducibility, fewer side effects ect.), together with high specificity and sensitivity.
Reviewer 2 Report
Major comments
The research question(s) of the study are not mentioned.
The study contains only patients with isolated GHD, known to at higher risk for normalization of GH results during transition period. This should be made clear in the title and the conclusion.
The studied population were tested around the age of 10.7 years, where in general priming before testing is needed to exclude false positive diagnosis of GHD. The lack of priming is a major shortcoming and should be discussed, as it is the major reason for normalisation of GH testing after pubertal development.
The studied population at the moment of testing should be better described ( height SDS corrected for parental height, growth velocity, birth weight SDS, IGF-1 SDS , pubertal status) to judge the severity of GHD or possibility of a constitutional delay. The name of the assays used should be given.
In most European countries the insulin tolerance test is used during transition, as the GHRH- arginine test might underdiagnose hypothalamic origins of GHD, which is the most frequent form of GHD in childhood. The GH to GHRH-arginine at the first testing and repeated testing should be given.
No mention of posterior pituitary ectopy in the MRI results, although this is a frequent finding in children with non acquired GHD. Was this anomaly studied ?
The results are poorly discussed and a more general discussion is given not taking the personal findings into account. The conclusion might well be that diagnostic criteria for GHD should be made more strict in stead of advocating earlier testing, where the study was not designed for.
Author Response
The research question(s) of the study are not mentioned.
o Added as requested.
The study contains only patients with isolated GHD, known to at higher risk for normalization of GH results during transition period. This should be made clear in the title and the conclusion.
o Modified as requested.
The studied population were tested around the age of 10.7 years, where in general priming before testing is needed to exclude false positive diagnosis of GHD. The lack of priming is a major shortcoming and should be discussed, as it is the major reason for normalisation of GH testing after pubertal development.
· The request was explained.
The studied population at the moment of testing should be better described ( height SDS corrected for parental height, growth velocity, birth weight SDS, IGF-1 SDS , pubertal status) to judge the severity of GHD or possibility of a constitutional delay. The name of the assays used should be given.
o We have added pubertal stage and difference in SDS between height and MPH. Currently we are not able to report IGF-1 SDS, birth weight SDS and growth velocity for all patients.
In most European countries the insulin tolerance test is used during transition, as the GHRH- arginine test might underdiagnose hypothalamic origins of GHD, which is the most frequent form of GHD in childhood. The GH to GHRH-arginine at the first testing and repeated testing should be given.
o We chose the GH-RH + Arginine test because it is more useful (reproducibility, fewer side effects ect.), together with high specificity and sensitivity.
No mention of posterior pituitary ectopy in the MRI results, although this is a frequent finding in children with non acquired GHD. Was this anomaly studied?
o Yes it was, but it was not found in any of our patients
The results are poorly discussed and a more general discussion is given not taking the personal findings into account. The conclusion might well be that diagnostic criteria for GHD should be made more strict instead of advocating earlier testing, where the study was not designed for.
o Modified as requested.
Reviewer 3 Report
This is a well written study by Penta et al that confirms that childhood GH deficiency is seldom permanent. I only have a minor comments:
In Materials and methods:
1. 2.1, 2nd paragraph, please add the mid-parental height (MPH) “in cm”
2. 2.1, page 3 first paragraph, would clarify the sentence “According to previous studies, we considered a small pituitary gland. . .” is not an exclusion
3. 2.1, please clarify how testicular volume was measured.
4. 2.2, please note whether or not the same assay and the same GH standard was used for all the patients, as that could change the GH responses
Results
5. Was the mean age at retesting different for males vs. females, as on average, males achieve adult height at a later age
6. This study uses a cut-off of<19 ng/mL and reports 5 patients with that response, however, the table reports 6 patients (numbers 4, 5, 13, 19, 25 and 28). It’s 5 patients if the criteria of Dreismann of 15.9 ng/mL is used.
7. State what the brain MRI alteration was for the one male patient of the 5 subjects with a pathologic GH peak
8. Last paragraph should be part of the preceding one, as they are both about lipid profile alterations
Table
9. In the table, please note what the abbreviation “Tp” stands for in the table legend (treatment period?); add IGF-I z-score
10. Reference ranges for IGF-I levels should be given
11. Since IGF-I concentrations were converted to SDS for age and sex, that should also be included in the table (and will be more meaningful than absolute levels, although I would keep those)
Discussion
12. Need to discuss that since the average age of GH therapy was 10.7 +/- 2.87 years, failure to GH stimulation testing could have been due to lack of sex steroid exposure (assuming these patients did not receive sex steroid priming)
13. Should note difference or not in result if Dreismann cut-off of 15.9 ng/mL is used vs. Italian cut-off
14. Bizzari study is quoted, but there are others – should give a range of results from various studies.
Author Response
In materials and methods
1. 2.1, 2nd paragraph, please add the mid-parental height (MPH) “in cm”
o Modified as requested.
2. 2.1, page 3 first paragraph, would clarify the sentence “According to previous studies, we considered a small pituitary gland. . .” is not an exclusion
o Modified as requested.
3. 2.1, please clarify how testicular volume was measured.
o Modified as requested.
4. 2.2, please note whether or not the same assay and the same GH standard was used for all the patients, as that could change the GH responses
o Yes, it was.
Results
5. Was the mean age at retesting different for males vs. females, as on average, males achieve adult height at a later age
o Mean age at retesting for males and females was added.
6. This study uses a cut-off of<19 ng/mL and reports 5 patients with that response, however, the table reports 6 patients (numbers 4, 5, 13, 19, 25 and 28). It’s 5 patients if the criteria of Dreismann of 15.9 ng/mL is used.
o Modified as requested.
7. State what the brain MRI alteration was for the one male patient of the 5 subjects with a pathologic GH peak
o Modified as requested, it was only a misprint.
8. Last paragraph should be part of the preceding one, as they are both about lipid profile alterations
o Modified as requested.
Table
9. In the table, please note what the abbreviation “Tp” stands for in the table legend (treatment period?); add IGF-I z-score
o Modified the abbreviation as requested. We are not able to report IGF-1 z-score
10. Reference ranges for IGF-I levels should be given
o Reference ranges are in ng/mL.
11. Since IGF-I concentrations were converted to SDS for age and sex, that should also be included in the table (and will be more meaningful than absolute levels, although I would keep those)
o We included data on -2SDS and + 2SDS.
Discussion
12. Need to discuss that since the average age of GH therapy was 10.7 +/- 2.87 years, failure to GH stimulation testing could have been due to lack of sex steroid exposure (assuming these patients did not receive sex steroid priming)
• The request was explained
13. Should note difference or not in result if Dreismann cut-off of 15.9 ng/mL is used vs. Italian cut-off
o Modified as requested.
14. Bizzari study is quoted, but there are others – should give a range of results from various studies.
o Modified as requested.